# Educational Methods and Technological Innovations for Introductory Experiential Learning Given the Contact-Related Limitations Imposed by the SARS-CoV2/COVID-19 Pandemic

**DOI:** 10.3390/pharmacy9010047

**Published:** 2021-02-25

**Authors:** Paul M. Reynolds, Erica Rhein, Monika Nuffer, Shaun E. Gleason

**Affiliations:** School of Pharmacy and Pharmaceutical Sciences, University of Colorado Skaggs, 12850 E Montview Blvd, Aurora, CO V20 1116R, USA; paul.reynolds@cuanschutz.edu (P.M.R.); Erica.rhein@cuanschutz.edu (E.R.); monika.nuffer@cuanschutz.edu (M.N.)

**Keywords:** patient simulation, education, pharmacy, telemedicine, educational technology

## Abstract

Background: Acute respiratory syndrome related coronavirus disease (COVID-19) has led to substantial changes in pharmacy curricula, including the ability to provide in-person introductory experiential practice experiences (IPPEs) to University of Colorado’s International-Trained PharmD (ITPD) students. Methods: The IPPE course for ITPD students was redesigned to offer remote educational activities in the health system setting and simulated practice and communication activities in the community setting. Students were evaluated via surveys regarding the perceived value of these changes, and changes in knowledge, skills and abilities before and after activities. Results: A total of 6 students were enrolled in the revised IPPE course. Students agreed or strongly agreed that the overall distance-based IPPE experience, the remote health system activities, and the community activities were valuable. Students also strongly agreed that course design successfully met course outcomes and was relevant to pharmacy practice. In terms of knowledge, skills and abilities, numeric improvements were observed in remote health system activities and community-based simulated patient interactions, but results were not statistically significant. A high baseline level of knowledge led to minimal improvements in perceptions of improvement in community pharmacy skills regarding pharmacy simulation software. Conclusion: Implementation of distance-based IPPE activities may be an alternate educational modality.

## 1. Introduction

The novel severe acute respiratory syndrome-related coronavirus disease (COVID-19, SARS-CoV-2) has infected over 84 million patients worldwide, leading to 1.8 million deaths at the time of the submission of this manuscript [1]. With a basic reproduction number ranging from 1 to 2.7 newly infected individuals for each infected patient with COVID-19 [2,3] and a global case fatality ranging from 1.7% to 8.8% [1,4], efforts have been implemented by global health organizations to curb the spread of the virus and subsequent saturation of health care resources. These efforts include social distancing, mask wearing, travel restrictions, increased testing and avoidance of large social gatherings [5]. 

As a result of the global pandemic, to date, substantial adjustments have been made by institutions conferring pharmacy education in order to continue operations and provide an effective pharmacy education while complying with global and national health organization guidance measures. Schools have had to adapt to changes regarding building operations, faculty gatherings, and fiscal policies [6]. 

At the same time, the transition to digital delivery of pharmacy education has forged new innovations as pharmacy educators are taking instructional risks by adopting new technologies, teaching methods, and assessment strategies in order to maintain student engagement and fulfill the educational mission [7]. Similar adjustments have been made by pharmacy experiential programs to meet the educational needs of these students due to health care rotation sites no longer permitting in-person visits in order to minimize risk of viral exposure and transmission to patients [8]. These have included shifting existing rotations to online formats, expansion of telemedicine, remote access to electronic health records (EHR), and remote precepting strategies. In response, similar approaches have been taken by other professions [9] and pharmacy schools on a global scale [10,11,12]. 

The University of Colorado Skaggs School of Pharmacy and Pharmaceutical Sciences (CU-SSPPS) offers two distance-based PharmD curricular pathways and one Masters in Clinical Pharmacy Program as part of its Distance Degrees and Programs. The International-Trained PharmD Program (ITPD) enrolls practicing pharmacists from around the world seeking to advance their careers and health care in their home country through the receipt of a PharmD degree [13]. Although the majority of the 90-credit hour curriculum is delivered in the distance-based medium, students travel to the school of pharmacy for 2 live (4 week) sessions. The first live session, which occurs upon entry to the program, introduces US pharmacy practice and exposes the student to the US health care setting through community and health care introductory pharmacy practice experiences (IPPEs). The second live session, which occurs upon completion of the didactic curriculum and prior to advanced pharmacy practice experience (APPE), integrates clinical skillsets learned from the classroom environment into direct patient care through an advanced-introductory pharmacy practice experience (aIPPE). The global pandemic caused by COVID-19 afforded unique challenges to this student population. At the time of this course implementation (July 2020), the 7 day average of new cases in Colorado was 475, and 65,000 new cases were being reported on a weekly basis [4]. Given this surge in cases, travel restrictions and mandatory quarantines were imposed by Colorado and a majority of the resident countries for the ITPD student population. This led to reduced availability of experiential practice sites, while still needing to meet the Accreditation Council for Pharmacy Education (ACPE) standard 12.6 of 300 clock hours of experience in the IPPE setting [14]. As a result, we were faced with the need to transform this session into the digital environment. 

In the summer (July–August) semester of 2020, changes were made to the content, structure, design, and implementation of the existing ITPD experiential program in order to accommodate learning needs in this international student population during the COVID-19 pandemic. The design and intent of these changes in the introductory experiential curriculum were to provide a similar educational experience without compromising intended educational outcomes or, ultimately, contributing to a substantial delay in graduation timelines. This paper describes these critical changes and compares the effectiveness of distance-based experiential learning strategies to in-person experiential rotations, and explores student perceptions of knowledge, skills, and abilities gained through this adapted curriculum. In addition, student performance and perceptions of the learning experience are explored. We discuss how these new learning strategies may be utilized moving forward for other PharmD programs. 

## 2. Materials and Methods 

### 2.1. Course Design

This prospective educational cohort study, using a mixed-methods research and educational approach, was approved by the Colorado Multiple Institutional Review Board (COMIRB). The IPPE course was a 2.5 semester credit hour offering to ITPD students newly enrolled in the program, starting in summer (July) of 2020. It also coincided with two didactic courses, US Pharmacy Practice Fundamentals (2.0 semester credit hours) and US Patient-Centered Communications (2.5 semester credit hours), which introduced the students to US-based pharmacy services, pharmacy calculations, formulary management, medication therapy management, reporting of medication errors, and patient assessment and communication. The IPPE course was offered online through the learning management system, Canvas (Instructure, Inc. Salt Lake City, UT, USA). Zoom (San Jose, CA, USA) videoconference software was also utilized for synchronous preceptor-student interactions. Orientation to the teaching methods and technology was provided to students within the learning management system, and in general welcoming informational Zoom videoconferences. Due to the COVID-19 pandemic, the first 60 h of IPPE activities were moved to remote (patients receiving care by telehealth) and simulated (health care software mimicking health systems environments and standardized patients) settings, as permitted by ACPE standard 12.7 [14], and contemporary COVID-19 guidance from ACPE. These 60 h were further divided into community and health system-related IPPE activities. 

For the health system IPPE setting, 20 h were allocated for a remote learning experience via the Zoom teleconferencing platform and the Epic electronic health record (EHR) system (Verona, WI, USA). After students completed Health Insurance Portability and Accountability Act (HIPAA) training, students were granted access to the EHR and the digital landscape of the hospital practice site, which included access to patient health records. As part of their first activity, students received a synchronous tour of the department of pharmacy at University of Colorado hospital by an author of this manuscript (PMR). A basic overview of dispensing operations, automated dispensing cabinets, medication storage, labeling and distribution was provided during the tour. Differences between centralized staffing and unit based clinical pharmacy staffing were outlined with the students. An overview of different pharmacy practice models in the health system setting [15] as well as potential clinical pharmacy specialist positions were provided to the students upon completion of the pharmacy tour. Students then provided a written reflection on which pharmacy practice models would be most effective in their home country, as well as what clinical pharmacy subspecialties were needed the most in their home country’s health care system. As students progressed through the course, they were provided access to patient charts from the internal medicine and critical care settings. Students used these charts to evaluate the effectiveness of collaborative practice agreements regarding pharmacy-directed dosage adjustments for renal dysfunction, identifying potential dosing changes given the dynamic changes in renal function these patients experience. Students were also required to complete an assignment of their own choice regarding the hospital’s local antibiogram. Students chose from selecting empiric coverage based on site of infection, tailoring of therapy to a specific pathogen, de-escalation of therapy based on stewardship principles, or dosage adjustment based on pharmacist-directed therapeutic drug monitoring (renal dosage adjustment or antibiotic levels). Both collaborative practice assignments were accompanied by a live Zoom reflection session with preceptors in which the details of clinical decision making surrounding these protocols were discussed further. The last health system assignment was a discussion of strategies utilized by clinical pharmacists to organize clinical information for rounding in a systematic manner. Students practiced “work ups” of patient charts using different clinical approaches in order to simulate preparation for rounding with a multidisciplinary team. This was facilitated with a further discussion of organizational strategies and prioritization of patients by clinical pharmacists providing service to these patient populations. 

For the community IPPE setting, 40 hours were allocated to the simulated learning pharmacy environment, MyDispense (Monash University, Melbourne, Australia), and communication activities related to prescription and non-prescription medication use in the community pharmacy setting. In order to introduce the community pharmacy setting in the US, the US version of MyDispense was used. This virtual pharmacy simulation software has been successfully used by other institutions for IPPE preparation, professional skills-building courses, pharmacy self-care encounters, pharmacy law, and as a personalized way to reflect on didactic education [16,17,18,19,20,21]. The software provides an overview of every aspect of the dispensing process, including prescription intake, product selection, drug interaction screening, product dispensing, product labeling, error resolution, interfacing with prescribers over a telephone system, and patient counseling. 

Students were able to choose from a total of 98 different activities in the MyDispense software program. Group “A” activities involved processes essential to dispensing legal and accurate prescriptions. In general, feedback was an automated comparison that provides a comparison of the student-generated final prescription product to an answer key with the correct prescription label. Feedback was added pertinent to both US- and Colorado-specific counseling laws as well as Naloxone standing orders, which permit pharmacist-directed provision of the reversal agent to patients receiving opioid prescriptions [22]. Group “B” activities required a higher order of clinical reasoning, with activities related to care of patients utilizing non-prescription remedies depending on the clinical context of the patient case. Group “C” activities required the highest order of clinical reasoning, often pertaining to the integration of drug-interaction software, patient vitals, laboratory values, patient questions pertaining to disease states, prior fill records, and literature pertaining to the disease states. These activities also covered unique situations in the community setting, such as mental health emergencies. Because these activities required more extensive input from the student, reflections were required in order for formative feedback to be provided by faculty members responsible for the activities. 

Activities were assigned different levels and point values on the basis of case complexity and difficulty. Students were eligible for up to 80 points of activities in the MyDispense program, with Group “A” activities counting for 1 point each, Group “B” counting for 2 points each, and Group “C” counting for 4 points each. A total of 80 Group A activities were made available to students (80 points), 10 Group B activities (20 points), and 8 group C activities (40 points). Students could elect for any combination of these activities to meet the assigned the 80 points of activities required for the course.

In order to reinforce skillsets surrounding collection of patient information and effective communication in the community IPPE setting, a variety of simulated activities were employed. Pre-recorded simulated virtual patient interviews (Skillful Productions), which were used previously by past cohorts later in the curriculum. Upon review, these activities were determined to be appropriate for this early curriculum course. These virtual patient interviews allowed students to gather appropriate information from a simulated patient, supporting content covered in the didactic communications course. Using branching logic, answers provided by the simulated patient varied based on the questions chosen by the student. In addition, feedback was provided about the appropriateness of the question posed to the simulated patient. Information gathered from this interview was then used by students in several different ways, including completion of a patient case worksheet (PCW), and recorded patient presentations to a preceptor and counseling appropriate for the simulated patient. These recordings were evaluated by faculty and constructive feedback was provided. 

Following the simulated virtual interviews, students were able to practice patient interviewing and communication skills in real time by interviewing a standardized patient. Students conducted two synchronous interviews for two standardized patients presenting with different disease states, which included scaly dermatoses and pediatric fever. After interviewing the standardized patient and gathering the relevant information, students completed a PCW and recorded a patient presentation. These activities were accompanied by written reflections for self-improvement, which asked students to identify aspects of the interview that went to their satisfaction as well as potential areas for improvement. Students were provided feedback by faculty regarding their performance on all of these tasks. 

As ACPE requires a total of 300 h of introductory pharmacy practice experiences, the remaining 240 h of IPPE activities have been planned for the students affected by the educational changes due to COVID-19. A total of 100 live (in-person) hours in the community setting will be required prior to completion of the didactic curriculum, accompanied by 20 h gained through the performance and reflections on 20 entrustable professional activities (EPAs) [23] directly supervised by a pharmacy preceptor. Of these EPAs, 10 of are repeated and refined, with the written reflections noting initial and refined learning. In addition, students will have to complete 120 h of health system-related activities during their advanced IPPE (aIPPE) rotation prior to qualifying for advanced pharmacy practice experience (APPE) rotations. This will be accompanied by reflective course work supervised by faculty members and preceptors. 

### 2.2. Course Assessment

A combination of formative and summative assessments were utilized for the remote and simulated learning activities in this IPPE course. Grade breakdowns in this course reflected the number of hours students dedicated to a respective activity. A total of 15% of the course grade was allocated to the remote pharmacy learning experiences in the health system setting, which consisted of attendance evaluations, preceptor evaluation of student professionalism and communication, and submission of the required assignments as outlined in the course design section of this manuscript. In addition, 29% of the course grade was allocated to community pharmacy activities, which included completion of simulated activities related to the MyDispense program and the communication activities in response to virtual and standardized patient interviews. Formative feedback was provided for completion of non-reflective (Group “A”) MyDispense activities. Summative feedback was provided for reflective (Group “B” and “C”) MyDispense activities, patient presentations, live interviews, PCWs, and reflections related to self-care and patient communication. The remaining 56% of the course grade is dedicated to future live hours (80 in total) spent in the community IPPE setting and will be a combination of attendance and preceptor evaluation of student performance.

### 2.3. Survey of Student Perceptions of IPPE Activities

Upon completion of the course, students completed a 28-question survey detailing baseline demographics regarding their pharmacy career (country of pharmacy degree, additional certifications, prior practice experience), level of agreement with the value of each educational experience, and confidence regarding specific knowledge and skills as a result of the remote health systems activities, simulated community IPPE activities, and communication activities before and after completion of this course. In addition, students were sent course evaluations regarding the quality, content, and ability of the course to meet intended outcomes. Responses to the survey remained optional and responses were anonymized. 

### 2.4. Statistical Analysis

Descriptive statistics were utilized to collate baseline demographics from survey data and academic performance in the course. A 4-point Likert scale (range: strongly disagree to strongly agree) was utilized to assess student perceptions of each learning experience. A 5-point scale compared student knowledge skills and abilities surrounding each educational method, technology, course outcomes, and faculty evaluations (1 = poor, 2 = fair, 3 = good, 4 = very good, 5 = excellent) both before and after the course. 

Survey consistency was assessed using Cronbach’s alpha coefficient. The Mann-Whitney U test was utilized to compare data pre- and post-course completion. *p* values of less than 0.05 were considered statistically significant.

## 3. Results

### 3.1. Student Demographics

Six ITPD students completed the IPPE course during the COVID-19 pandemic, of which five completed the survey regarding the novel teaching methods outlined for this course. Of these students, two received their originating pharmacy degrees from India, one from Sudan, one from Lebanon, and one from the Philippines. Four students had a bachelor’s degree in pharmacy as the highest degree prior to entry to the program and one had a master’s degree prior to the program. All survey respondents had retail or community pharmacy experience prior to program entry, with a median of 6 years of experience (IQR 3 to 8.5 years). Two survey respondents had prior health system pharmacy experience, with a median of 8.5 years (IQR 2 to 15 years) of experience. 

### 3.2. Student Assessment of Pedagogical Value

Student assessments of different pedagogical approaches in this course are outlined in Table 1.

Overall, students highly valued the distance-based IPPE experience implemented. Median scores suggest strong agreement that the course in its redesigned form conferred a positive learning experience. Average responses regarding remote health system activities demonstrated that students strongly agreed that learning experiences from this section of the course were valuable to future pharmacy practice. However, a majority of students exhibited strong agreement that they would prefer a live activity over remote health system activities. Similar results were exhibited when student impressions of the value of MyDispense were assessed. Scores ranged from agree to strongly agree regarding the value of the program for increasing confidence in the community pharmacy setting, understanding of the prescription dispensing process in the US, and overall value to learning. Similar to the remote health system IPPE experiences, students preferred live activities over MyDispense. The use of pre-recorded virtual patients and live standardized patients were valued (strongly agree for all categories) by students for the purpose of building communication skills, practicing effective patient counseling techniques, and preparation for patient presentations in future courses. Cronbach’s alpha coefficient for survey data was 0.94, suggesting a high degree of internal consistency.

All six students completed the course evaluation. Students selected agree or strongly agree for all questions regarding the ability of the course design to meet outcomes, perceptions of active learning strategies, course organization, variety of learning strategies to stimulate learning, and the courses relevance to the practice of pharmacy in Table 2. 

Narrative comments are optional on course evaluations and subsequently information provided by students was limited. A common theme identified was the request for increased communication including the provision of clearer instructions regarding expectations and estimated workload for various activities related to the course. 

### 3.3. Perceived Growth in Knowledge, Skills, and Abilities

Student perceptions regarding growth in knowledge, skills, and abilities as a result of the remote IPPE activities offered are outlined in Table 3. 

Students found themselves improving from good to either very good or excellent for collecting data via the EHR, organizing patient data, developing a care plan, monitoring progress, presenting patient information concisely, and answering drug inpatient information questions. However, these results were not statistically significant. Statistically significant improvements in communication were observed (progression from very good to excellent). Students observed a slight decline in professionalism (excellent to very good) but this result was not statistically significant. Confidence in clinical abilities in the health system setting remained the same both before and after the rotation (very good). Student assessment of knowledge, skills and abilities before and after MyDispense use did not numerically or statistically increase, with the exception of monitoring patient progress (progression from good to very good). Of note, students rated themselves as very good to excellent in these skillsets at baseline before MyDispense use, which remained unchanged after MyDispense use. Virtual and standardized patient communication content improved students’ abilities to verbally present and clinically document findings gained from patient interviews, but these results were not statistically significant.

### 3.4. Academic Performance

At the time of the writing of this manuscript, all ITPD students have completed the distance-based portion of the course with a mean academic performance of 98% (± SD of 0.7). Average performance on IPPE health system activities was 98% and performance on communication related to standardized/virtual patients was 97.9%. All students completed the 80 required points for the MyDispense program (receiving 100% for this portion of the course). All 6 students completed the least complex “A” level activities for an average of 70 points per students, 5 students completed the higher complexity “B” level activities for an average of 13 points per student, and 2 students completed “C” level activities for an average of 8.67 points per student. In terms of progression in the remainder of IPPE activities, all students still have remaining live IPPE hours to complete before qualifying for aIPPE and APPE rotations. Two students have completed 1 out of the 20 required entrustable professional activities (EPAS). One student completed an EPA regarding counseling a patient prescribed a new antibiotic, which was rated at a skill level of 4 by a preceptor (intermittently supervised). The other EPA was related to engaging in interprofessional practice, which was rated at a skill level of 3 (reactive supervision by a preceptor). Both of these EPAs are relevant to skillsets conferred in this course.

## 4. Discussion

The rapid changes made to the ITPD IPPE course were successful in allowing students to complete a substantial proportion of experiential education in a distance-based setting. Changes made to this IPPE course allowed students to begin to progress through the program while gaining valuable exposure to US pharmacy practice in both the health system and community settings. Due to travel restrictions, it was impossible or logistically infeasible for students to travel internationally to attend a live session. Therefore, this combination of remote and simulated experiences remained the only viable option to provide these students with an introduction to the experiential setting. Although these educational methods were well received by the students taking this new iteration of the IPPE course, it remains feasible that this positive reception was a product of the circumstances of the global pandemic as opposed to the overall educational value of the course. However, data collected from students in the form of standard course evaluations and a 28-question survey, both anonymous, demonstrate that they considered the learning experience provided to be positive and resulted in self-reported progression of key skills. Academic performance also aligned with these results. 

Other studies have detailed adaptations to experiential and didactic pharmacy curriculum due to COVID-19 [9,10,11,12]. Changes to experiential curriculum previously described include a shift to virtual or remote rotations, use of simulation or supplemental activities, rearrangement of rotation schedules, and changes to rotation structure to decrease risk to students [9,10,11,12]. Our study is unique from several standpoints in comparison to these works. Although these works have focused on the adaptation of curricular strategies to effectively address educational needs during COVID-19, no study to date has assessed student perceptions of these changes beyond the impact of COVID-19 on pharmacy student well-being [24]. This study adds to the literature surrounding this subject by providing measures of the effectiveness of a multimodal distance-based educational approach to experiential activities affected by the pandemic. Our educational approach differed from that offered in Australia, which offered flexibility by either offering student deferment of live rotations due to health concerns or continuation of live experiential activities despite the pandemic [11]. Our separate approach was largely due to limitations in rotation availability from US practice sites during the pandemic eliminating any possibility of early live rotations. The approach described by pharmacy schools in Saudi Arabia was similar to ours in terms of describing a virtual dispensing platform to IPPE students, but was limited in scope as there was only a 1-week cessation of experiential activities due to the pandemic in this country [10]. Simulated experiences have been adopted by other health care professions, including nursing education. In a randomized trial, nursing students who had up to 50% of clinical hours replaced by simulation had no significant differences in any preceptor ratings (including student communication or clinical critical thinking) compared to students who received clinical education solely in the live setting [25]. These data suggest that as a general modality, distance-based simulated and remote educational strategies may not only supplement, but also may potentially replace components of in-person experiential learning activities. As COVID-19 has led to a paradigm shift in pharmacy education, the above strategies should be considered for future student needs surrounding accommodations, refugee status restricting travel abroad, or other potential ethical and practical situations restricting a student’s ability to complete all experiential activities in the live setting [9]. 

Several limitations to this study have been identified, many of which are specific to this cohort and circumstances. With the rapidity of the onset of COVID-19, survey data were not prospectively validated prior to survey implementation which may subject the survey to question bias and may minimize overall validity. However, the Cronbach’s alpha coefficient did suggest the survey had a high degree of internal validity. Due to the very small sample size, obtaining statistically significant results was not as feasible as with a larger student cohort. However, notable trends in the data suggested improvements in key skillsets and statistical analysis suggested a high degree of internal consistency in survey responses. 

All three activity categories aligned with patient care domains of the Center for Advancement of Pharmacy Education (CAPE) 2013 Outcomes, Domain 2 Essentials for Practice and Care and Domain 3 Approach to Patient Care, and two to Domain 4 Personal and Professional Development, as outlined in Table 4 [26]. Our assessments, however, were not broken down by CAPE outcome. This will be important to note in future assessments of these methods, especially in regards to performance on simulated or remote patients versus live, in-person patients.

Students progressed along an increasingly challenging trajectory for the live health system and patient communication facets of this course. In contrast, lack of perceived improvement from MyDispense activities may be explained by a variety of factors, including self-directed learning. Unlike previous studies regarding MyDispense, all students enrolled in this course had prior community pharmacy experience in their respective home countries, resulting in a high level of baseline confidence in pharmacy practice-based skillsets demonstrated by the software. In addition, as students were allowed to direct their learning, a majority of the activities completed in MyDispense were in the less-challenging groups of activities. This is possibly due to student workload, adaptation to the online environment, familiarization with the software, or a variety of other factors. The resultant pursuit of less-challenging activities in MyDispense may have created a higher degree of perceived confidence regarding pharmacy practice activities captured by this program. 

As this cohort was new to the program, they did not have previous experience with the structure of the program or the educational technologies utilized. While we provided written and brief verbal information on the methods used, more extensive orientation to educational technologies and expectations could potentially have been beneficial. As the MyDispense software was new to faculty members in this program prior to this iteration of the course, improvements could be made to the design, flow, and feedback from activities related to this software. Further data are needed in order to draw a comparison between distance-based educational methods for experiential activities and live educational activities. As students enrolled in this course progress through the program, further assessments will be provided to discern their perceptions regarding their abilities to perform live activities in IPPEs and the role of the distance-based IPPE in their preparation for APPEs and clinical practice. For programs considering implementation of similar educational technologies, the following recommendations should be considered through the implications of these research results. First, MyDispense represents a growing collaborative network of academic pharmacists who have formulated cases, content, and questions in the program. Under the pretext of active contribution to this community, this program continues to grow and flourish as schools participate in contributions on an international level. Offering students a menu of choices, but asking them to complete more advanced activities would further stratify and challenge their capabilities for growth. Second, students enjoyed activities surrounding direct application of learning to real patient cases. This level of immersion provided tangibility and relevance to the patient care processes and communication skills conveyed in our doctorate of pharmacy curriculum. Lastly, further steps will need to be undertaken to evaluate the effectiveness of these strategies as the pandemic continues and society continues to be transformed by technology and telehealth. 

## 5. Conclusions

This study shows an effective use of remote and simulated activities to provide international pharmacy students with an introduction to US-based community and health system clinical environments. Future studies are needed regarding the comparison of remote and simulated experiential activities with live activities in health care education. These changes and innovations may be necessary in order for pharmacy education to remain robust in light of future challenges beyond COVID-19. 

## Figures and Tables

**Table 1 pharmacy-09-00047-t001:** Perceived Value of IPPE Topic or Pedagogy by Student Responders (*n* = 5).

Topic or Pedagogy	Value (Median, IQR) *
Overall IPPE Experience during COVID-19	4 (3.5–4)
**Remote Health System Activities**	
Hospital Tour, Dispensing Operations, Practice Models	4 (3.5–4)
Health System Collaborative Practice Activities (renal dosage adjustment, antibiotic tailoring)	4 (3.5–4)
Preference for Live Activity over Remote Health System Activities	4 (2.5–4)
Ability to Apply Knowledge Gained from Health Systems Activities	3.5 (3–4)
Remote Health System Activities Should Be Offered in the Future	4 (2–4)
**Simulated MyDispense Community Pharmacy Activities**	
MyDispense Improved Understanding of Rx Dispensing in the US	3 (1.25–4)
Increased Student Confidence in Community Setting	4 (2–4)
Preference for MyDispense over a Live Activity	2 (1–3.5)
Usefulness of Feedback from MyDispense	4 (2–4)
Realism of Simulated Interface	3 (2.5–4)
Overall Value to Learning	3 (1.5–4)
Ability to Apply Knowledge Gained from MyDispense	4 (3–4)
Mydispense Should Be Offered in the Future	3 (1.5–4)
Value of Group “A” Activities	3 (1.5–4)
Value of Group “B” Activities	No student completed
Value of Group “C” Activities	No student completed
**Remote and Simulated IPPE Non-Prescription Communication Activities**	
Presentations and interviews built on concepts and skills from other courses	4 (2.5–4)
Preparation to use presentation skills in other courses	4 (3–4)
Simulated patients provided structure for interview questions and information gathering that can be utilized in the live setting	4 (3–4)
Simulated patient interviews prepared the student well for the standardized patient interview	4 (3–4)
Ability to practice communication with a standardized patient	4 (2.5–4)
Recorded counseling and patient presentations-built communication confidence	4 (3–4)

* 4-point Likert scale (1 = strongly disagree, 2 = disagree, 3 = agree, 4 = strongly agree). Abbreviations: IPPE, introductory pharmacy practice experience; IQR, interquartile range.

**Table 2 pharmacy-09-00047-t002:** Student Course Evaluations Related to the Distance-Based IPPE Course (*n* = 6).

Question	Median Response *
Course was designed to meet outcomes	5
Course helped student meet expectations for professional behavior	5
Active learning and lab-based activities helped meet course outcomes	5
Course was well organized	4.5
Variety of learning strategies were offered to stimulate student learning	5
Course was made relevant to practice of pharmacy	5
IPPE preceptors and faculty helped student to achieve course goals and objectives	5

* 5-point Likert scale (1 = strongly disagree, 2 = disagree, 3 = neither agree nor disagree, 4 = agree, 5 = strongly agree). Abbreviations: IPPE, introductory pharmacy practice experience.

**Table 3 pharmacy-09-00047-t003:** Student Perceptions of Change in Knowledge, Skills and Abilities from Distance-Based IPPE Activities.

	Before Experience	After Experience	*p*-Value
**Change in Perceptions of Knowledge Skills and Abilities—Remote Health System IPPE Activities**			
Collecting data via EHR	3	5	0.19
Organizing patient data	3	5	0.23
Developing or modifying a care plan based on changes to a collaborative practice agreement (renal or antibiotic adjustment)	3	4	0.11
Monitoring patient progress	3	5	0.15
Presenting patient information concisely	3	5	0.10
Answering drug information questions relevant to inpatient care	4	5	0.12
Navigating the hospital drug information system	1	4	0.31
Communicating with Peers, faculty, other health care professionals, and patients	4	5	0.048
Demonstrating professionalism and trustworthiness	5	4	1.0
Confidence in clinical abilities in the health system environment	4	4	0.27
**Change in Perceptions of Knowledge Skills and Abilities—Simulated through MyDispense**			
Rx label preparation	5	5	0.79
Analyzing a prescription	5	5	0.9
Identifying medication errors	4	5	0.18
Monitoring patient progress	3	4	0.5
Using drug information resources	4	4	0.74
Accepting new Rx via phone	5	5	0.9
Rx transfer	5	5	0.43
Analyzing patient medication profiles	5	4	1.0
Accurately composing a medication label	5	5	0.51
Drug selection from inventory	5	5	0.8
Application of appropriate warning labels	4	4	0.5
Counseling of a patient	4	4	0.57
Intervening on an incorrect order	4	4	0.729
Overall confidence in community environment	5	5	0.69
**Change in Perceptions of Knowledge Skills and Abilities—Remote and Simulated Standardized Patient Interactions**			
Collection of patient information	4	4	0.25
Verbal presentation of patient health information to a preceptor	3	4	0.12
Clinical documentation of findings	3	4	0.23

* 5-point Likert scale (1 = strongly disagree, 2 = disagree, 3 = agree, 4 = strongly agree). Abbreviations: IPPE, introductory pharmacy practice experience; EHR, electronic health record; Rx, prescription.

**Table 4 pharmacy-09-00047-t004:** CAPE * Outcomes Addressed in This Course.

Activity	Corresponding CAPE * Outcome
**Remote Health System IPPE**	
Pharmacy tour, pharmacy practice models, reflection on desired practice and optimal model for home country	2.2.1; 2.2.2; 2.2.3; 4.1.3
Collaborative practice agreements (renal dosing, antibiotic adjustment, patient work-up scenarios)	1.1; 1.1.6; 2.1.1; 2.1.2; 2.1.3; 2.1.4; 2.1.5; 2.1.6; 2.1.7; 2.2.6; 3.1; 3.6.9
**Simulated and Standardized Patient Interviews**	
Simulated patient interview, counseling	2.1.1; 3.3.1; 3.6.1; 3.6.2; 3.6.3
Standardized patient interaction	2.1.1; 3.1; 3.3.1; 3.6.1;3.6.2; 3.6.3; 3.6.4; 3.6.5; 3.6.6; 3.6.7; 3.6.8; 3.6.9; 4.4.1
**MyDispense**	
Level “A” activities	2.1.1; 2.1.7; 2.2.1; 2.2.3; 2.3.2; 2.3.3; 3.6.9
Level “B” activities	2.1.1; 2.1.7; 2.2.1; 2.2.3; 2.3.2; 2.3.3; 3.1.1; 3.1.3; 3.1.4; 3.6.9
Level “C” activities	2.1.1; 2.1.7; 2.2.1; 2.23; 2.3.2; 2.3.3; 3.1.1; 3.1.1; 3.1.3; 3.1.4; 3.1.5; 3.1.6; 3.6.9

* Center for Advancement of Pharmacy Education (CAPE).

## Data Availability

Survey data related to this study are available upon request from the corresponding author.

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
