# Peer review of "Educational Methods and Technological Innovations for Introductory Experiential Learning Given the Contact-Related Limitations Imposed by the SARS-CoV2/COVID-19 Pandemic"

_pharmacy, 2021, doi:10.3390/pharmacy9010047_

Round 1

Reviewer 1 Report

The article assesses the effect of remote implementation in-person introductory experiential practice experiences of IPPE at a distance. The results of the study among students show that remote IPPE was an effective educational method during COVID-19.
The topic seems interesting due to the many changes in science and technology development during the COVID-19 pandemic. The introduction of this method has been described in detail, although I am curious if there were any difficulties in introducing this service? I think it is worth describing. Additionally, please consider the following suggestions:
- indication of the COVID-19 data in the region where the study was conducted (and not only in the world)
- review of similar solutions in the US and around the world (e.g. in discussion)
- indication of "good practices" / recommendations of such implementations, to give the article a practical dimension.

Reviewer 2 Report

Abstract:  In the  Conclusion section- suggestion to re-phrase the concluding sentence since the extrapolation of the results is over-rated.

Method: was the survey carried out psychometrically tested prior to implementation particularly to test validity, reliability and reduce interview-question bias?

Discussion:  Effectiveness was tested from a student perspective point of view and very limited with respect to CAPE educational outcomes measurement especially with real patients- maybe this should be reflected more in the discussion chapter
